# Structural mapping of Na$_v$1.7 antagonists

Qiurong Wu[1,7], Jian Huang[2,7] ✉, Xiao Fan[2,7] ✉, Kan Wang[3,7], Xueqin Jin[1,7], Gaoxingyu Huang[4,5], Jiaao Li[1], Xiaojing Pan[1] ✉ & Nieng Yan [1,2,6] ✉

Voltage-gated sodium (Na$_v$) channels are targeted by a number of widely used and investigational drugs for the treatment of epilepsy, arrhythmia, pain, and other disorders. Despite recent advances in structural elucidation of Na$_v$ channels, the binding mode of most Na$_v$-targeting drugs remains unknown. Here we report high-resolution cryo-EM structures of human Na$_v$1.7 treated with drugs and lead compounds with representative chemical backbones at resolutions of 2.6-3.2 Å. A binding site beneath the intracellular gate (site BIG) accommodates carbamazepine, bupivacaine, and lacosamide. Unexpectedly, a second molecule of lacosamide plugs into the selectivity filter from the central cavity. Fenestrations are popular sites for various state-dependent drugs. We show that vinpocetine, a synthetic derivative of a vinca alkaloid, and hardwickiic acid, a natural product with antinociceptive effect, bind to the III-IV fenestration, while vixotrigine, an analgesic candidate, penetrates the IV-I fenestration of the pore domain. Our results permit building a 3D structural map for known drug-binding sites on Na$_v$ channels summarized from the present and previous structures.

Voltage-gated sodium (Na$_v$) channels govern membrane excitability in neurons and muscles[1,2]. Nine subtypes of Na$_v$ channels, Na$_v$1.1-Na$_v$1.9, are responsible for firing electrical signals with tissue specificity[3]. Consistent with their fundamental physiological significance, aberrant activities of Na$_v$ channels are linked to a wide range of pathogenic conditions. Na$_v$ channels are directly targeted by a number of FDA-approved anti-epilepsy drugs (AEDs), painkillers, and antiarrhythmic agents[4–8].

High-resolution (cryo-EM) structures of eukaryotic Na$_v$ channels reported in recent years offered an advanced understanding of the mode of action (MOA) of some of the drugs at atomic level. For instance, the antiarrhythmic drugs, flecainide, quinidine, and propafenone are all accommodated in the central cavity of the pore domain (PD) of Na$_v$1.5, but with non-overlapping binding poses[9–11]. Bulleyaconitine A (BLA), an active ingredient isolated from *Aconitum* plant for pain management and for the treatment of rheumatoid arthritis in

China, cuts off the ion permeation by sitting right below the selectivity filter (SF) in the structure of Na$_v$1.3[12]. Nonetheless, the accurate binding mode of most of the Na$_v$-targeting drugs is yet to be unveiled. Such information will not only reveal the chemical basis for the MOA of the clinically applied drugs, but also lay an important foundation for future drug development[13].

We sought to identify the druggable sites on Na$_v$ channels by carrying out a systematic structural investigation using human Na$_v$1.7-β1-β2 as the scaffold[14,15]. Na$_v$1.7, encoded by *SCN9A* and primarily localized to the dorsal root ganglion neurons, has been explored as a prominent target for pain management[16]. Loss of function of Na$_v$1.7 leads to indifference to pain[17–19], whereas mutations that enhance channel activities are found in patients with primary erythromelalgia, paroxysmal extreme pain disorder and small fiber neuropathy[20–23]. Despite extensive efforts from a number of preeminent pharmaceutical companies, most of the Na$_v$1.7-targeting drug candidates failed to

[1]Beijing Frontier Research Center for Biological Structures, Tsinghua-Peking Joint Center for Life Sciences, School of Life Sciences, Tsinghua University, Beijing 100084, China. [2]Department of Molecular Biology, Princeton University, Princeton, NJ 08544, USA. [3]Department of Anesthesiology, China-Japan Friendship Hospital, Beijing 100029, China. [4]Westlake Laboratory of Life Sciences and Biomedicine, Key Laboratory of Structural Biology of Zhejiang Province, School of Life Sciences, Westlake University, 18 Shilongshan Road, Hangzhou 310024 Zhejiang Province, China. [5]Institute of Biology, Westlake Institute for Advanced Study, 18 Shilongshan Road, Hangzhou 310024 Zhejiang Province, China. [6]Shenzhen Medical Academy of Research and Translation, Guangming District, Shenzhen 518107 Guangdong Province, China. [7]These authors contributed equally: Qiurong Wu, Jian Huang, Xiao Fan, Kan Wang, Xueqin Jin. ✉e-mail: jh6493@princeton.edu; xiaof@princeton.edu; panxj@tsinghua.edu.cn; nyan@tsinghua.edu.cn

meet the endpoint(s) of phase II trials[8,24–26]. We reckoned that structural details of $Na_v1.7$ in complex with clinically applied drugs and lead compounds may afford a molecular insight that will benefit rational drug design or/and optimization.

In the structures of eukaryotic $Na_v$ channels, the core α subunit, which generally consists of 1500-2200 residues, is well resolved for the transmembrane (TM) and extracellular regions[12,15,27–34]. The four 6-TM (S1-S6) containing repeats exhibit a pseudo symmetry[33,35,36]. The S1-S4 segments in each repeat constitute the flanking voltage-sensing domain (VSD), and the S5-S6 segments enclose the PD that is responsible for selective and gated ion permeation[28,33,34,37]. Above the PD, several heavily glycosylated extracellular loops (ECL) are stabilized by multiple disulfide bonds[32,38].

To explore potential druggable sites, we selected representative drugs and lead compounds with diverse chemical backbones (Supplementary Table 5). Our efforts were unfortunately limited by the availability of the chemicals. Here we present cryo-EM structural analysis of $Na_v1.7$ treated with the following drugs: a local anesthetic drug bupivacaine (Marcaine) and a chemically related AED lacosamide (Vimpat), an anticonvulsant drug carbamazepine (Tegretol) that is also used for the treatment of trigeminal neuralgia and epilepsy[39–41], a synthetic vincamine derivative vinpocetine (Cavinton) that can target $Na_v$ channels for the treatment of cerebrovascular disorders, a natural product hardwickiic acid isolated from plant *Croton californicus* with antinociceptive effect, and vixotrigine (also known as raxatrigine), which is a state- and use-dependent $Na_v1.7$ blocker under clinical investigation for ameliorating trigeminal pain and neuropathic pain[42–44]. Our present structural studies, along with previous reports, reveal that the PD is a versatile receptor for ligands with distinct chemical formulas and structures.

## Results

### Cryo-EM analysis of $Na_v1.7$ with different drugs

We started with more drugs than the abovementioned ones, including a local anesthetic lidocaine, an analgesic under development funapide (XEN402)[45], a pain killer candidate DSP-2230[8], and a selective $Na_v1.7$ inhibitor PF-05089771 that targets $VSD_{IV}$[46,47]. To validate the function of some purchased antagonists, we performed preliminary whole-cell patch-clamp electrophysiological recordings in HEK293T cells (Supplementary Figs. 1–2 and Supplementary Tables 1–2). Based on these characterizations, the antagonists were incubated with purified $Na_v1.7$-β1-β2 complex for 1 h at a final concentration of at least 10-fold higher than their measured $IC_{50}$ values before cryo-sample preparation.

Following our reported cryo-EM data acquisition and processing protocols[14,33], we obtained ten 3D EM reconstructions for the $Na_v1.7$-β1-β2 complex treated with different small molecules at overall resolutions of 2.6–3.2 Å (Supplementary Figs. 3–4, Supplementary Figs. 8–10 and Supplementary Table 3). Cross comparison of 3D EM maps immediately reveals unambiguously resolved densities for bupivacaine (BPV), lacosamide (LCM), carbamazepine (CBZ), vinpocetine (VPC), hardwickiic acid (HDA), vixotrigine (VXT), and PF-05089771. However, no density is found for lidocaine, and the local resolution is insufficient to unambiguously assign funapide or DSP-2230, even when these compounds were added throughout the entire purification procedure during repeated attempt to resolve the ligands (Supplementary Table 5).

The lack of density for lidocaine and limited resolutions for funapide and DSP-2230 may result from various reasons. Detergents, which were applied at high concentrations, may interfere with the drug binding. In fact, we sought to solve the structure of human $Na_v1.5$ bound to lidocaine with multiple attempts, but never observed any ligand density. It is thus not surprising that no lidocaine is found in the EM map for $Na_v1.7$ either. Structures of both $Na_v1.5$ and $Na_v1.7$ display similar inactivated state. It is also possible that some ligands may

prefer different conformations that are yet to be resolved. In the following, we will discuss the well resolved ligands only.

Recognition of PF-05089771 by $Na_v1.7$-$VSD_{IV}$ was previously examined on a $Na_vAb$ chimera where the extracellular half of the VSD is replaced by the counterpart in $Na_v1.7$-$VSD_{IV}$[47]. Our incentive to solve the structure of PF-05089771 bound to wild-type (WT) human $Na_v1.7$ was to verify if the chimeric VSD can faithfully recapitulate all binding details. The binding mode of PF-05089771 in the intact $Na_v1.7$-$VSD_{IV}$ is remarkably similar to the chimeric one (Supplementary Fig. 5). As coordination details and the molecular basis for the subtype-specificity have been thoroughly discussed by Ahuja and co-workers, we will not elaborate on this one. In the following, we will focus on drug binding to the PD.

### A common site for BPV, LCM, and CBZ

In the 3D EM map of all WT human $Na_v$ channels, the intracellular gate is penetrated by a linear density that likely belongs to the detergent molecule glyco-diosgenin (GDN) or digitonin. In the maps of $Na_v1.7$ treated with BPV, LCM, and CBZ, the linear density is replaced by shorter ones with distinct features that respectively match the three drugs (Fig. 1a). Instead of wedging through the gate, these densities block it from the intracellular side. Therefore, BPV, LCM, and CBZ occupy the same site beneath the intracellular gate, which we name as site BIG. Unexpectedly, LCM has a second binding pose that directly plugs into the selectivity filter from the central cavity (Fig. 1b, c). We will discuss the second site in a later session. For illustration simplicity, we will name LCM as LCM-1 and LCM-2 in site BIG and the central cavity, respectively.

Analysis of the ion permeation path using HOLE[48] shows that the intracellular gate is substantially narrowed (Fig. 1d). The radius of the constriction site falls to ~1 Å, which is more than 1 Å shorter than that in the apo channel (PDB code: 7W9K)[14]. Gate contraction results from an α→π transition in the middle of the $S6_{IV}$ segment, in addition to minor inward movements of $S6_{II}$ and $S6_{III}$ (Fig. 1d). Similar conformational changes have been observed for $Na_v1.7$ upon binding to the peptide toxins Protoxin-II (ProTx-II) or Huwentoxin-IV (HWTX-IV)[14]. ProTx-II binds to the extracellular side of $VSD_{II}$ and $VSD_{IV}$, which are respectively known as Site 4 and Site 3 for neurotoxin binding to $Na_v$ channels; HWTX-IV recognizes Site 3[15,49,50]. It is interesting that peptide toxins and small molecule drugs that bind to unrelated sites lead to similar gate rearrangement.

In apo-$Na_v1.7$, the intracellular gate is of an oval contour constituted by Leu398, Leu960, Phe963, Ile1453, Val1752, and Tyr1755 from the four S6 segments. In the presence of the drugs BPV, LCM, or CBZ, the three S6 segments other than $S6_I$ all slightly move toward the center in addition to the α→π transition of $S6_{IV}$. Consequently, four gating residues, Leu398, Leu964, Ile1457, and Ile1756, move closer to tighten the gate (Fig. 1d). It is noted that Leu964, Ile1457, and Ile1756 are all engaged in the coordination of the three drugs (Fig. 2). Such conformational shifts are thus coupled to site BIG formation.

### Site BIG

Site BIG is constituted by the cytosolic residues on the S6 tetrahelical bundle that are adjacent to the gating residues. Although the three drugs completely overlap in site BIG, the coordinating details vary. BPV is primarily surrounded by hydrophobic residues, including Leu964, Leu967, Leu968, and Phe971 from $S6_{II}$, Ile1457 from $S6_{III}$, and Ile1756 and Leu1760 from $S6_{IV}$ (Fig. 2a). The 3,5-dimethylphenyl and the piperidine rings of BPV, linked by an amide group, align on a similar plane against the BIG site, leaving the butyl tail pointing to the cytoplasmic side.

The purchased BPV is a racemate of dextro-(*R*)- and levo-(*S*)-bupivacaine. To investigate if BPV binding has racemic specificity, we performed *in-silico* docking simulations. The aromatic ring and

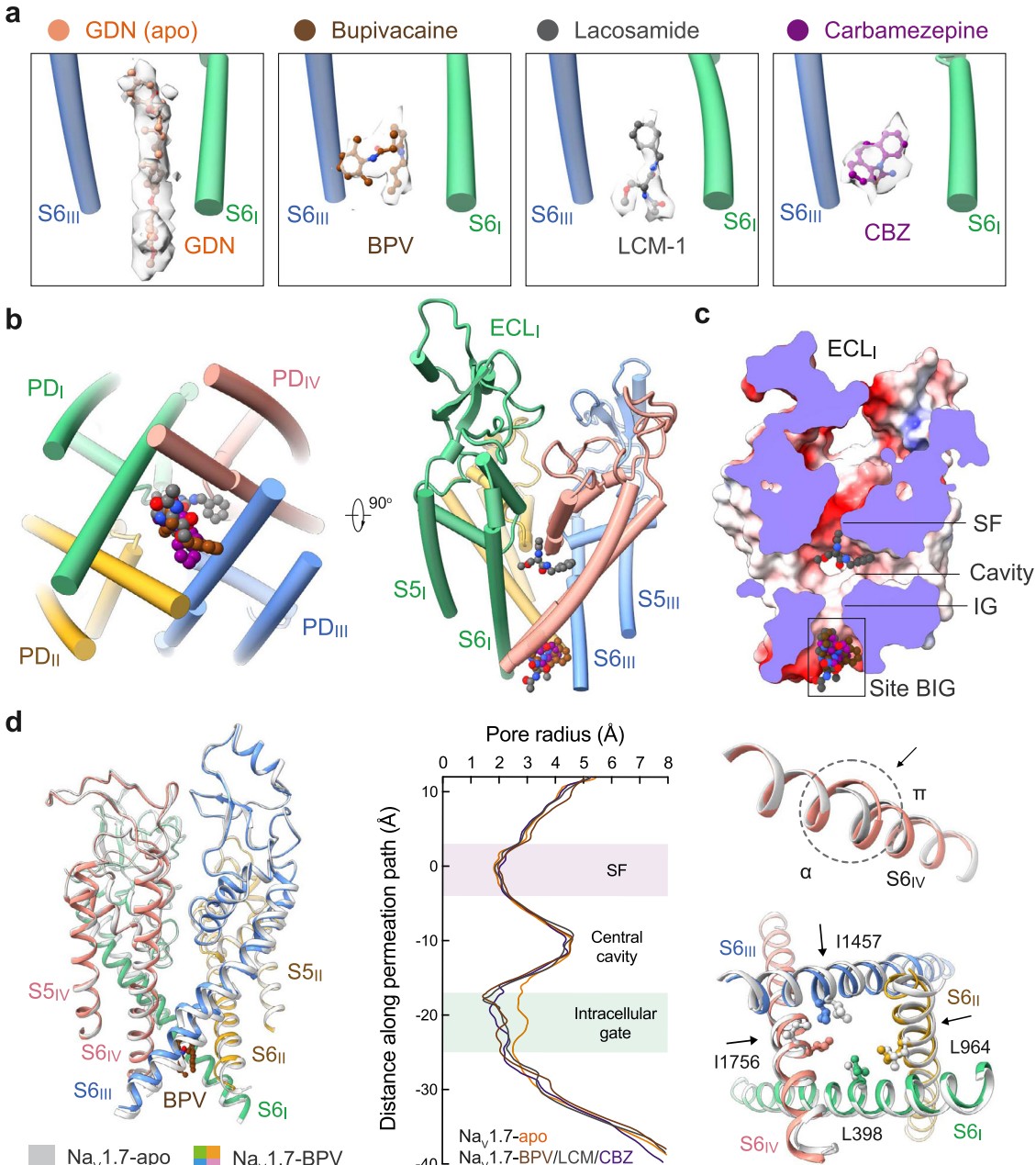

**Fig. 1 | BPV, LCM, and CBZ bind to the same site beneath the intracellular gate (site BIG). a** Distinct densities for the three drugs at site BIG. From *left* to *right*, the chemical structures (ball and sticks) and corresponding densities (semi-transparent cloud) are shown for GDN (Na$_V$1.7-apo, PDB code: 7W9K), bupivacaine (BPV, brown), lacosamide (LCM-1, grey), and carbamezepine (CBZ, purple). The densities for the small molecules are presented at similar levels in Chimera, 5 σ for GDN, 4 σ for BPV, 4 σ for LCM-1, and 4.5 σ for CBZ. For visual clarity, only part of the S6$_I$ and S6$_{III}$ segments are shown. **b** The three drugs overlay at Site BIG. A bottom view (*left*) and a side view (*right*) of the superimposed pore domain (PD, domain colored) with bound drugs (BPV, LCM, and CBZ) are shown. The same color scheme for the four repeats is applied throughout the manuscript. **c** Drugs at site BIG directly block the intracellular gate. Shown here is a cross-section view of ligand-bound Na$_V$1.7 PD in an electrostatic surface representation. Two distinct binding poses are observed for LCM (grey spheres), one at site BIG (LCM-1) and the other below the SF (LCM-2). **d** Further contracted intracellular gate in the presence of the site BIG-binding drugs. *Left*: Structural comparison of Na$_V$1.7-BPV and Na$_V$1.7-apo (silver). *Middle*: The pore radii of Na$_V$1.7 bound to different ligands are calculated in HOLE and tabulated. *Right*: The α→π transition of S6$_{IV}$ and rearrangement of the gating residues, such as Leu398, Leu964, Ile1457 and Ile1756 (shown as ball and sticks), in the presence of BPV lead to gate contraction. LCM and CBZ-bound structures display nearly identical conformations to that of Na$_V$1.7-BPV.

the piperidine basic nitrogen of these two isomers largely overlap in site BIG. The primary difference in the binding poses of the isomers is the opposite orientations of the butyl tail (Supplementary Fig. 6). Therefore, both racemic isomers of BPV can be recognized at site BIG.

LCM-1 and CBZ occupy the same cavity of site BIG as BPV, but form additional hydrogen bonds (H-bonds) (Fig. 2b, c). The amide

nitrogen from the central amide group of LCM-1 is H-bonded with the carboxyl group of Glu406 on S6$_I$ (Fig. 2b), and the carbonyl oxygen of CBZ receives an H-bond from Asn1461 on S6$_{III}$ (Fig. 2c). The three clinical drugs share a conserved binding mode, wherein the aromatic rings project into the center of the pore, leaving the flexible tail to the intracellular side. Drugs with the same chemical skeleton may share a similar binding pose.

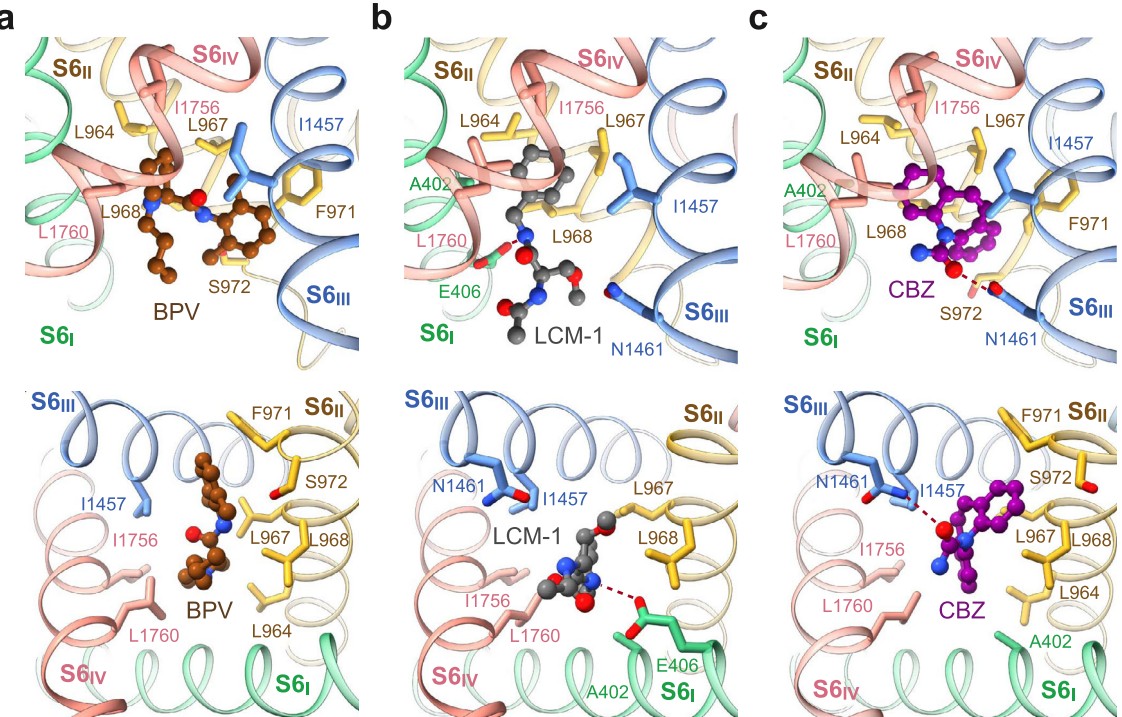

**Fig. 2 | Drug coordination at the BIG site.** Binding details of (**a**) BPV, (**b**) LCM-1, and (**c**) CBZ at site BIG are presented in identical side (*top*) and intracellular views (*bottom*). The drugs are shown as ball and sticks, and surrounding residues are shown as sticks. The potential hydrogen bonds are indicated with red dashed lines.

## LCM-2 directly blocks the SF

The central cavity of voltage-gated ion channels (VGICs) is a versatile platform for various chemicals. Eukaryotic Na$_v$ and Ca$_v$ channels comprise four non-identical repeats, thereby offering more docking sites than homo-tetrameric VGICs.

Beneath the ceiling of the central cavity, the Y-shaped LCM-2 is positioned off the central axis, closer to repeats I and IV (Fig. 3a, b). The benzyl ring is embedded in a pocket enclosed by residues Ser1697, Ile1744, Phe1748, Val1751, and Val1752. The two amides are H-bonded to the backbone carbonyl group of Thr1696 on the P1 loop of repeat IV, forming a stable anchor for LCM-2. The carbonyl oxygen from the acetylamino moiety interacts with Lys1406, which is one constituent of the DEKA motif on repeat III, and the amide nitrogen engages in an H-bond with Gln360, which is stabilized by the side chain hydroxyl group of Ser390 (Fig. 3c). The extensive H-bond network may account for the stronger density of LCM-2 than that of LCM-1 (Figs. 1a, 3a).

## Vinpocetine and hardwickiic acid bind to the III–IV fenestration

The III–IV fenestration in the L-type Ca$_v$ channels is a well-known site for dihydropyridine compounds[13, 51, 52]. The corresponding site in Na$_v$ channels has been suggested to be targeted by local anesthetics[53–57]. Although no density for lidocaine is found around this site, densities for VPC and HDA were resolved[40,58]. A battery of hydrophobic residues on S6$_{III}$ and S6$_{IV}$ are engaged in accommodating VPC or HDA, including Thr1404 on P1$_{III}$, Trp1332 on S5$_{III}$, Thr1448, Leu1449, and Phe1452 on S6$_{III}$, and Ser1697, Ile1744 and Phe1748 in repeat IV (Fig. 4a–c). Notably, the binding pose for VPC, which extends deeper into the central cavity, only partially overlaps with that of HDA in the III–IV fenestration (Fig. 4a).

## Vixotrigine penetrates the IV–I fenestration

A stretch of elongated density is observed traversing the IV-I fenestration in VXT-treated Na$_v$1.7. Linear density with similar pose is often found to belong to a lipid tail in apo-Na$_v$ channels. Fortunately, the bulges that are characteristic of the three ring moieties facilitated model building of VXT (Fig. 4a, b). VXT is held by a number of

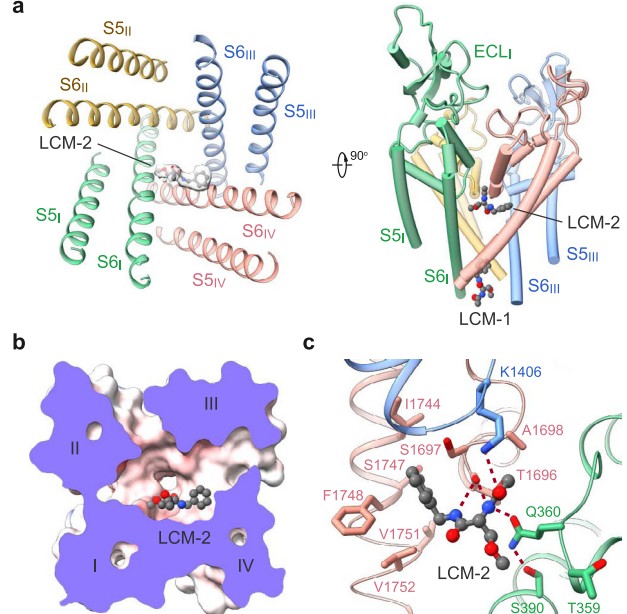

**Fig. 3 | Lacosamide binds to two distinct sites. a** Accommodation of lacosamide at the second binding site in the central cavity. The contour level for LCM-2 in the left panel is 5 σ. **b** A cut-open electrostatic surface of Na$_v$1.7-LCM-2 viewed from the extracellular side. **c** Coordination of LCM-2 in the central cavity. Surrounding residues are shown as sticks. The potential hydrogen bonds are indicated with red dashed lines.

hydrophobic residues on S6$_I$ and S6$_{IV}$, including Phe1692, Thr1695 and Thr1696 on P1$_{IV}$, Ile386, Phe387 and Phe391 on S6$_I$, and Val1751 and Tyr1755 on S6$_{IV}$. The amido group from pyrrolidine moiety and carbonyl oxygen are H-bonded to the phenolic hydroxyl group of Tyr1755 on S6$_{IV}$, forming a stable anchor for VXT (Fig. 4c).

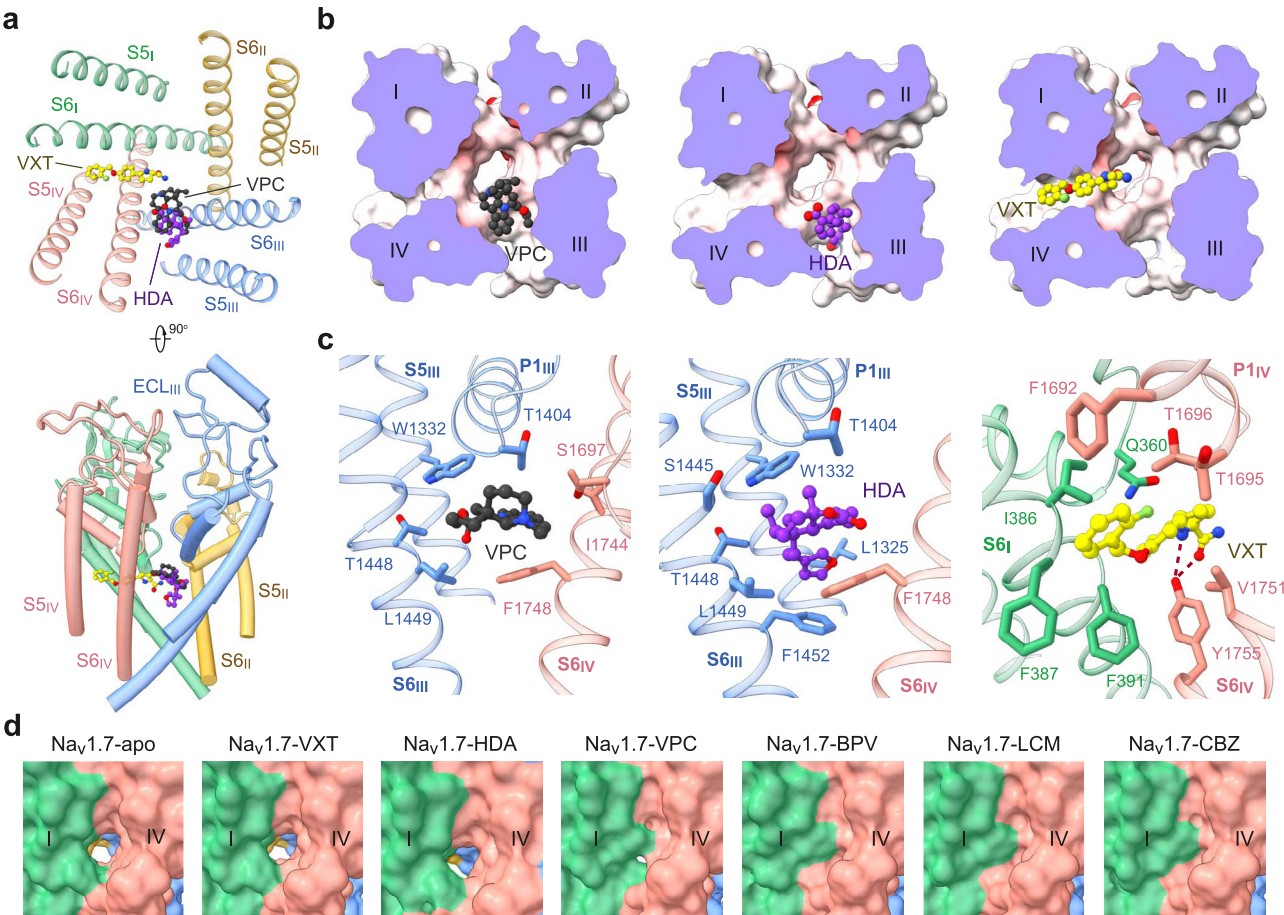

**Fig. 4 | Versatile fenestration binding sites on the PD. a** VPC and HDA are accommodated at the III-IV fenestration, and the elongated linear VXT projects into the central cavity from the IV-I fenestration. Two perpendicular views of the PD are shown. **b** Different binding poses at the fenestration sites. Shown here are cut-open electrostatic surfaces viewed from the extracellular side. The drugs are shown as spheres. **c** Details of VPC, HDA, and VXT coordination. **d** Different states of the IV-I fenestration in the presence of distinct pore blockers. The fenestration, which exists in the apo or VXT, HDA and VPC-bound channel, is gone in the presence of BPV, CBZ, or LCM accompanying the α→π transition of S6IV.

Unlike VPC, the majority of which is within the cavity (Fig. 4b), VXT tunnels through the IV-I fenestration, with its acylamino moiety below the SF and its benzyl ring contacting the lipid bilayer (Fig. 4a, b). It is noted that the IV-I fenestration is sensitive to the α→π transition of S6IV (Fig. 4d). In the π form, the fenestration is gone. Such binding pose thus immediately provides a molecular interpretation for the dependence on channel states for VXT binding.

## Discussion

### A structural map for drug binding sites on Nav channels

Nav channels are well-known targets for natural toxins and clinically applied drugs. Seven sites (Site 1 – Site 7) have been mapped to the primary sequence of Nav channels based on decades of rigorous functional characterizations[4] (Fig. 5a). With the growing number of high-resolution structures of Nav channels in the presence of toxins and drugs, the conventional definition of binding sites may no longer be able to reflect the diversity and complexity of the drug binding sites, particularly those on the PD. Hereby we map the known ligand binding sites to the resolved structures of Nav channels and propose a letter-coded nomenclature system to depict these druggable sites in the 3D space.

The conventional Site 3 and Site 4 refer to the extracellular segments of VSDIV and VSDII, respectively (Fig. 5a). Whereas small molecules like PF-05089771 resides in an extracellular cavity of VSDIV, peptide toxins like ProTx-II and HWTX-IV bind to the extracellular periphery of the VSD, partly immersed in the lipid bilayer[14,] [15,59–61]. Some other peptide toxins, such as Dc1a and AaH2, employ the adjacent pore segments as scaffold to secure the interaction with the VSD[62–64]. To distinguish these different binding poses on VSDs, we propose to name the binding sites as following: VnEC for the extracellular cavity (n = 1, 2, 3, 4, referring to repeats I, II, III and IV, respectively), VnEM and VnEP for the extracellular site involving the membrane and the pore domain, respectively. Per this definition, PF-05089771 binds to site V4EC, HWTX-IV and LqhIII to V4EM, and Dc1a to site V2EP (Fig. 5b). There is no known ligand that binds to the intracellular side of any VSD. If such binder is discovered or engineered in the future, the corresponding accommodation site can be named similarly as VnIC/M/P, where I stand for intracellular.

The pore domain hosts remarkably diverse ligand binding sites, in part owing to the asymmetry of the four repeats. The dome enclosed by the disulfide bond-stabilized ECLs represents an important receptor site for peptide toxins like μ-conotoxin KIIIA[28]. To acknowledge the contribution of ECLs from all repeats instead of the previously defined Site 1b, which refers to the residues on ECLII, we suggest naming this region site E (Fig. 5c).

Below the ECLs, the outer mouth to the SF vestibule is notoriously known as Site 1 or Site 1a that is targeted by the prototypical neurotoxins tetrodotoxin and saxitoxin[15,33,62]. We propose to name it Site S, standing for selectivity filter. Along the permeation path, the binding sites between the SF and site BIG may be named as Site C for the cavity and Site G for the gate (Fig. 5c). As the central cavity is

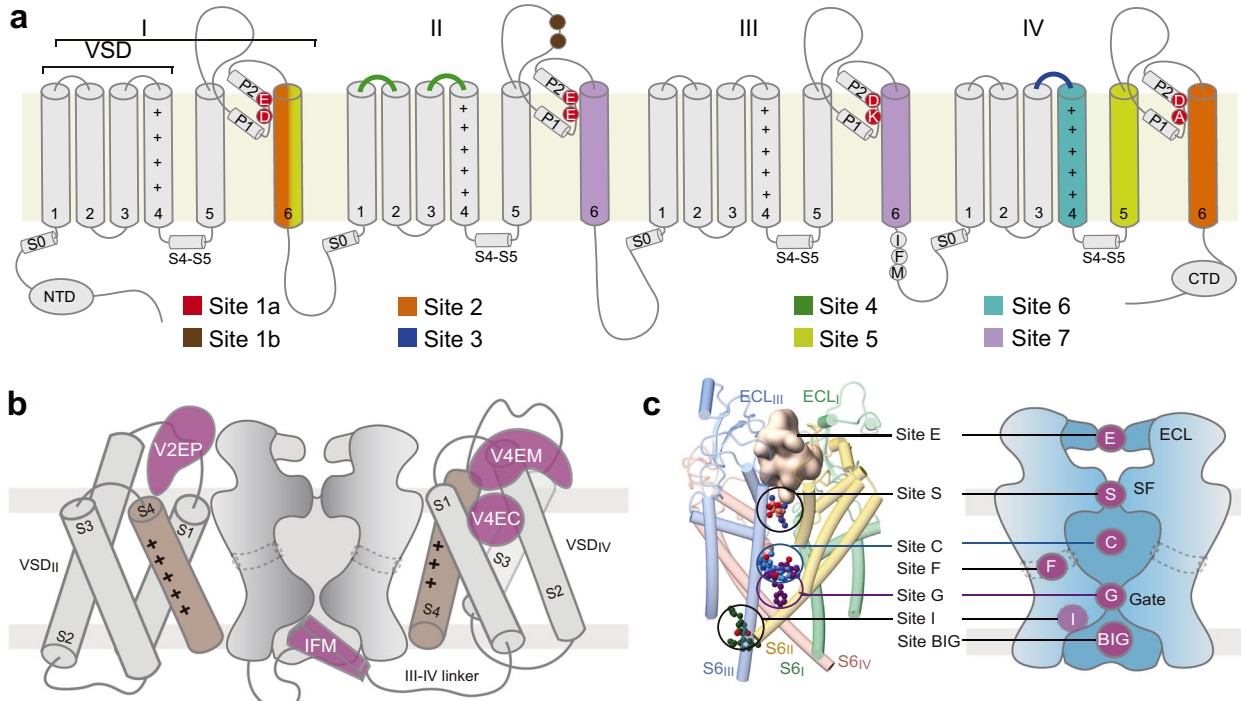

**Fig. 5 | Structural mapping of binding sites for Na$_v$ antagonists. a** Conventional mapping of drug and toxin-binding sites on the primary sequence of Na$_v$ channels. **b** Structural mapping of VSD sites for accommodating peptidic and small molecule ligands. Based on the new nomenclature system, these sites are named as VnEC/M/P. V: VSD; n: the repeat number; E: extracellular; C: cavity. M: membrane; P: pore domain. **c** The PD hosts multiple drug-binding sites. *Left*: Representative Na$_v$ inhibitors are mapped to the structural model of the PD. The peptide toxin μ-

conotoxin KIIIA is shown as a pale pink surface that is encaged by the extracellular loops (Site E). TTX is shown as orange spheres at the entrance to the SF (Site S). Quinidine, propafenone and cannabidiol are shown as blue, purple and green spheres that bind to the cavity (Site C), the intracellular gate (Site G) and the inactivation site (Site I)[65], respectively. A diagram of the druggable sites on the PD are shown on the right. Site F refers to the fenestration sites.

rather spacious for a small molecule, Site C can be further specified as UC, CC, and LC for upper cavity, central cavity, and lower cavity, respectively. Based on this classification, LCM-2 binds to site UC. It is noted that the bulk of LCM-2 is off the central axis, residing closer to repeats I and IV. The specific site can be referred to as UC$_{14}$ when the asymmetry needs to be highlighted. We also propose to use letter F to refer to the fenestration sites, such as Site F1 for the I-II fenestration and F4 for IV-I. Thus, bulleyaconitine A, vinpocetine and vixotrigine bind to Sites F1, F3 and F4, respectively. In the accompanying paper, we describe the action of cannabidiol (CBD) on Na$_v$1.7. Among the two binding poses of CBD, one involves the cavity where the fast inactivation motif is accommodated, and thus named site I[65] (Fig. 5c).

### Implications on drug discovery

The nine subtypes of Na$_v$ channels, owing to their tissue-specific distributions and unique biophysical properties, are preferentially associated with different disorders. Nevertheless, these channels share a high degree of sequence similarity, posing a major challenge for the development of subtype-specific drugs. Sites C, G, BIG, and F all involve the S6 segments, which are highly conserved (Supplementary Fig. 7). It is therefore unsurprising that many drugs that bind to these sites are non-selective Na$_v$ blockers. Nevertheless, there are subtype-specific inhibitors for Site C. For example, A-803467, which binds to Site CC$_4$, is a Na$_v$1.8-selective blocker[66]. The subtype specificity is conferred by remote allosteric loci, adding another tier of complications to drug design[31].

The S5 and S6 segments below the SF exhibit marked structural shifts associated with their functional states. For instance, the fenestrations may exist only in the activated and inactivated states, but not the resting state. The volume of the central cavity fluctuates

accordingly[67]. Furthermore, the axial rotation of the S6 segments accompanying the transition between the α and π helical turns results in reposition of S6 residues, reshaping the interior of the cavity. In addition, fenestrations and intracellular gate may be related to the entrance and exit pathways of small molecules. Consequently, many drugs that bind to sites C, F, G, and BIG are state-dependent. Such complexity necessitates structural resolution of the channels in different functional states.

Site V4C has been extensively explored for subtype-specific antagonists, such as PF-05089771 for Na$_v$1.7 and ICA121431 for Na$_v$1.3. An overlooked region is site E, which exhibits lower sequence conservation among Na$_v$ channels. Supporting this notion, KIIIA shows a strong subtype-selective sensitivity[28,68]. Site E may deserve more attention for the development of biologics that are of therapeutic potentials.

The development of non-addictive analgesics represents an unmet healthcare need. For a long time, rational design targeting Na$_v$ channels was impeded by the lack of accurate structures. Despite recent structure boom of Na$_v$ channels, AI-facilitated drug discovery is still hindered by the limitation of the training database wherein the number of ligand-bound structures remains scarce. Moreover, the conformational dynamics is beyond the current capacity of structure prediction. Therefore, experimental structures of target proteins bound to ligands are indispensable for drug discovery. Structural analyses reported here provide an important foundation to aid rational design of drugs targeting Na$_v$ channels.

## Methods

### Transient expression and purification of the human Na$_v$1.7-β1-β2 channel complex

The plasmids for human Na$_v$1.7, β1 and β2 are the same as reported previously[15]. HEK293F cells (Invitrogen) were cultured in SMM 293T-II

medium (Sino Biological Inc.) under 5% $CO_2$ in a Multitron-Pro shaker (Infors, 130 r.p.m.) at 37 °C, and transfected with plasmids when the cell density reached $2.0 \times 10^6$ cells per ml. For 1 liter cell culture, approximately 2.5 mg plasmids (1.5 mg for $Na_v1.7$ plus 0.5 mg for β1 and 0.5 mg for β2) were pre-incubated with 4 mg 25-kDa linear poly-ethylenimines (PEIs, Polysciences) in 25 mL fresh medium for 20 min before adding into cell culture. Transfected cells were cultured for 48 h before harvesting.

Protein purification was carried out following our optimized protocol[15]. 20 L cells co-transfected with $Na_v1.7$-β1-β2 were used for one batch of sample preparation. They were harvested by centrifugation at 800 g and resuspended in the lysis buffer containing 25 mM Tris-HCl (pH 7.5) and 150 mM NaCl. The suspension was supplemented with 1% (w/v) n-dodecyl-β-D-maltopyranoside (DDM, Anatrace), 0.1% (w/v) cholesteryl hemisuccinate Tris salt (CHS, Anatrace), and protease inhibitor cocktail containing 2 mM phenylmethylsulfonyl fluoride (PMSF), aprotinin (6.5 μg/mL), pepstatin (3.5 μg/mL), and leupeptin (25 μg/mL). For channels treated with carbamazepine or lacosamide, 1 mM drug molecule was supplemented throughout the purification procedure, while for channels treated with hardwickiic acid, 10 μM drug molecule was added throughout the whole purification process. After incubation at 4 °C for 2 h, the cell lysate was ultracentrifuged at 20,000 g for 45 min, and the supernatant was applied to anti-Flag M2 affinity gel (Sigma) by gravity at 4 °C. The resin was rinsed four times with the wash buffer (buffer W) containing 25 mM Tris-HCl (pH 7.5), 150 mM NaCl, 0.05% glycol-diosgenin (GDN, Anatrace), and the protease inhibitor cocktail. Proteins were eluted with buffer W plus 200 μg/mL FLAG peptide (Sigma). The eluent was then applied to Strep-Tactin Sepharose (IBA) and the purification protocol was similar to the previous steps except that the elution buffer was buffer W plus 2.5 mM D-Desthiobiotin (IBA). It was then concentrated using a 100-kDa cut-off Centricon (Millipore) and further purified through Superose-6 column (GE Healthcare) in buffer W. The peak fractions were pooled and concentrated to ~1 mg/mL, and further incubated with the small molecule drugs for 1 h before cryo-EM sample preparation. Additional carbamezapine (2 mM) and lacosamide (3 mM) were supplemented, making the final concentrations for small molecule drugs to 50 μM for vixotrigine and vinpocetine, 230 μM for bupivacaine, 3 mM for carbamazepine, 4 mM for lacosamide, 10 μM for hardwickiic acid, and 50 μM for PF-05089771. Lacosamide was purchased from Cayman Chemical, and other drugs were purchased from MedChemExpress.

## Whole cell electrophysiology

For whole-cell patch-clamp recordings, HEK293T cells (Invitrogen) were cultured in Dulbecco's Modified Eagle Medium (DMEM, BI) supplemented with 4.5 mg/mL glucose and 10% (v/v) fetal bovine serum (FBS, BI). Cells for recordings were plated onto glass coverslips and transiently co-transfected with the expression plasmid for $Na_v1.7$ and an eGFP-encoding plasmid using lipofectamine 2000 (Invitrogen). Cells with green fluorescence were selected for patch-clamp recording 18–36 h after transfection. All experiments were performed at room temperature. No further authentication was performed for the commercially available cell line. Mycoplasma contamination was not tested.

The whole-cell $Na^+$ currents were recorded using an EPC10-USB amplifier with Patchmaster software v2*90.2 (HEKA Elektronic), filtered at 3 kHz (low-pass Bessel filter) and sampled at 50 kHz. The borosilicate pipettes (Sutter Instrument) had a resistance of 2–4 MΩ and the electrodes were filled with the internal solution composed of (in mM) 105 CsF, 40 CsCl, 10 NaCl, 10 EGTA, 10 Hepes, pH 7.4 with CsOH. The bath solutions contained (in mM): 140 NaCl, 4 KCl, 10 Hepes, 10 D-glucose, 1 $MgCl_2$, 1.5 $CaCl_2$, pH 7.4 with NaOH. Data were analyzed using Origin (OriginLab) and GraphPad Prism (GraphPad Software).

To determine inhibition of the inactivated state by PF-05089771, cells were clamped at −120 mV for 200 ms before stepping to −70 mV (for $Na_v1.7$) or −80 mV (for $Na_v1.5$) for 5 s in order to get about 50% channel inactivated. This was followed by a 5 ms return to −120 mV preceding a 50 ms test pulse to 0 mV. To investigate the resting state blockage of $Na_v1.7$ by the small molecules, cells were held at −120 mV and stepped to +0 mV for 50 ms. Drugs were dissolved in dimethyl sulfoxide (DMSO, final concentration in the perfusion buffer is less than 0.1%). Solutions with indicated concentrations were freshly prepared and perfused to the recording cell for up to 10 mins to get to the maximal block using a multichannel perfusion system (VM8, ALA). Prior to drugs perfusion, cells were recorded for 5 min to establish stable peak current. Concentration-response curve was fitted with: $Y = Bottom + (Top − Bottom) / (1 + w10^{((LogIC_{50}-X) * Hill Slope)})$, where $IC_{50}$ is the concentration of small drugs that blocks 50% of the current and X is log of indicated concentration, and Hill Slope is slope factor. All data points are presented as mean ± SEM and n is the number of experimental cells from which recordings were obtained. Statistical significance was assessed using one-way ANOVA analysis and extra sum-of-squares F test.

## Cryo-EM data acquisition

Aliquots of 3.5 μL freshly purified $Na_v1.7$ with small drugs were placed on glow-discharged holey carbon grids (Quantifoil Au 400 mesh, R1.2/1.3). Grids were blotted for 3.0 s and plunge frozen in liquid ethane cooled by liquid nitrogen with Vitrobot Mark IV (Thermo Fisher). For vixotrigine/vinpocetine complex, electron micrographs were acquired on a Titan Krios electron microscope (Thermo Fisher) operating at 300 kV and equipped with a Gatan K3 Summit detector and GIF Quantum energy filter. A total of 5280/4534 movie stacks for $Na_v1.7$ bound to vixotrigine/vinpocetine were automatically collected using AutoEMation[69] with a slit width of 20 eV on the energy filter and a preset defocus ranging from −1.8 μm to −1.5 μm in super-resolution mode at a nominal magnification of 81,000. Each stack was exposed for 2.56 s with 0.08 s per frame, resulting in 32 frames per stack. For hardwickiic acid complex, electron micrographs were acquired on a Titan Krios electron microscope (Thermo Fisher) operating at 300 kV and equipped with a Gatan K3 Summit detector and GIF Quantum energy filter. A total of 6389 movie stacks for $Na_v1.7$ bound to hardwickiic acid were automatically collected using AutoEMation[69] with a slit width of 20 eV on the energy filter and a preset defocus ranging from −1.8 μm to −1.5 μm in super-resolution mode at a nominal magnification of 64,000. Each stack was exposed for 2.56 s with 0.08 s per frame, resulting in 32 frames per stack. For the carbamezapine/lacosamide complex, electron micrographs were acquired on a Titan Krios electron microscope (Thermo Fisher) operating at 300 kV and equipped with a Cs corrector, Gatan K2 Summit detector and GIF Quantum energy filter. A total of 4254/4268 movie stacks for $Na_v1.7$ bound to carbamezapine/lacosamide were automatically collected using SerialEM[70] with a slit width of 20 eV on the energy filter and a preset defocus ranging from −1.6 μm to −1.2 μm in super-resolution mode at a nominal magnification of 105,000. Each stack was exposed for 5.6 s with 0.175 s per frame, resulting in 32 frames per stack. For the PF-05089771/bupivacaine complex, electron micrographs were acquired on a Titan Krios electron microscope (Thermo Fisher) operating at 300 kV and equipped with Gatan K3 Summit detector and GIF Quantum energy filter. 6106/7361 movie stacks of $Na_v1.7$ bound to PF-05089771/bupivacaine were automatically collected using EPU (Thermo Fisher) with a slit width of 20 eV on the energy filter and a preset defocus ranging from −1.8 μm to −1.5 μm in super-resolution mode at a nominal magnification of 81,000. Each stack was exposed for 2.56 s with 0.08 s per frame, resulting in 32 frames per stack. The total dose was 50 $e^-/Å^2$ for all the datasets. The stacks were motion-corrected with MotionCor2[71] or Warp[72] and binned 2-fold, resulting in 1.0825 Å/pixel for vinpocetine and vixotrigine, resulting in 1.0979 Å/

pixel for hardwickiic acid, 1.114 Å/pixel for carbamezapine/lacosamide and 0.865 Å/pixel for PF-05089771/bupivacaine. Dose weighting was performed as described[73]. Defocus values were estimated using Gctf[74] or cryoSPARC[75].

## Image processing

For simplicity, the data processing of PF-05089771 was selected as representative with a diagram showed in Supplementary Fig. 3. A total of 7,991,055 particles were automatically picked using template picking in cryoSPARC. After 2D classification, a total of 551,861 good particles were selected, these particles were subject to one round of non-uniform refinement in cryoSPARC[76], resulting in 3D reconstructions at 2.8 Å. The resulting particles were subject to one additional round of Heterogeneous Refinement in cryoSPARC, with the number of classes set to 6. A total of 492,421 particles were selected and applied to another round of Non-uniform Refinement and CTF refinement, resulting in final reconstructions at overall resolutions of 2.7 Å (Supplementary Fig. 4 and Supplementary Table 3). Other datasets were processed with the same workflow.

## Model building and structure refinement

Model building was performed based on the map acquired in the data processing. The starting model of Na$_v$1.7 (PDB 7W9K), was fitted into the EM map by Chimera[77]. All Na$_v$1.7 residues, lipids, and sugar moieties were manually checked in COOT[78]. The chemical properties of amino acids were taken into consideration during model building.

Structural refinement was performed using phenix.real_space_refine application in PHENIX[79] with secondary structure and geometry restraints. Over-fitting of the overall model was monitored by refining the model in one of the two independent maps from the gold-standard refinement approach and testing the refined model against the other map[80]. Statistics of the map reconstruction and model refinement can be found in Supplementary Table 3.

## In silico molecular docking

After removing BPV from the Na$_v$1.7-BPV complex, both dextro-(*R*)- and levo-(*S*)-bupivacaine were docked against Na$_v$1.7 using Schrödinger Suite 2018-1 (Schrödinger, Inc.) to investigate the racemic specificity of BPV on Na$_v$1.7. The initial small molecule structures were generated and optimized using LigPrep program[81] with the OPLS3 force field[82], while the protein structure was processed using the default setting within Protein Preparation Wizard with the coordinates of the Na$_v$1.7-BPV complex as input. Molecular docking was performed using the Glide program with the extra-precision docking method (Glide XP).

## Reporting summary

Further information on research design is available in the Nature Portfolio Reporting Summary linked to this article.

## Data availability

The data that support this study are available from the corresponding authors upon request. The cryo-EM maps have been deposited in the Electron Microscopy Data Bank (EMDB) under accession codes EMD-35193 (Na$_v$1.7-BPV), EMD-40238 (Na$_v$1.7-LCM), EMD-40239 (Na$_v$1.7-CBZ), EMD-35197 (Na$_v$1.7-VPC), EMD-35975 (Na$_v$1.7-HDA), EMD-35198 (Na$_v$1.7-VXT), and EMD-35194 (Na$_v$1.7-PF-05089771). Coordinates have been deposited in the Protein Data Bank under accession codes 8I5B (Na$_v$1.7-BPV), 8S9B (Na$_v$1.7-LCM), 8S9C (Na$_v$1.7-CBZ), 8I5X (Na$_v$1.7-VPC), 8J4F (Na$_v$1.7-HDA), 8I5Y (Na$_v$1.7-VXT), and 8I5G (Na$_v$1.7-PF-05089771). The sequences of human Na$_v$1.7, β1 and β2 are available in the following links: Na$_v$1.7 (UniProtID:Q15858); β1 (UniProtID:Q07699); β2 (UniProtID:O60939).

Previously solved structures mentioned in this study are under the accession codes in PDB: 7W9K; 6J8J; 6J8I; 6J8E; 7W9M; 7W9T; 7W9P; 5EK0; 7XMF; 7XMG; 7XM9; 6LQA; 7FBS; 8G1A.

The source data underlying Fig. 1d, Supplementary Fig. 2a–b and 4 are provided as a Source Data file. Source data are provided with this paper.

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

## Acknowledgements

We thank Xiaomin Li, Fan Yang, and Jianlin Lei for technical support during EM image acquisition. We thank the Tsinghua University Branch of China National Center for Protein Sciences (Beijing) for providing the cryo-EM facility support. We thank the computational facility support from Tsinghua University Branch of China National Center for Protein Sciences (Beijing). We also thank the cryo-EM facility at Princeton Imaging and Analysis Centre. This work was funded by National Natural Science Foundation of China (32271252). X.F. has been supported by the HFSP long-term fellowship (LT000754) from the International Human Frontier Science Program Organization (HFSPO). N.Y. had been supported by the Shirley M. Tilghman endowed professorship from Princeton University in 2017–2022.

## Author contributions

Q.W., H.J., X.F., K.W. and J.L. prepared the sample. Q.W., J.H. and X.F. collected the EM data. Q.W., X.F. and G.H. analyzed the EM data and calculated the EM map. X.F., G.H. and X.P. built and refined the atomic model. X.J. designed and performed the electrophysiological experiments. Q.W., H.J., X.F., K.W., X.J., G.H. discussed and analyzed the results. Q.W., H.J., X.F., X.P and N.Y.analyzed the structure and wrote the paper.

## Competing interests

The authors declare no competing interests.
