## [Peer Review File · Nature Communications]

Structural mapping of Nav1.7 antagonistsReviewers' Comments:

Reviewer #1:

Remarks to the Author:

Manuscript by Qiurong Wu et al. describes cryoEM structures of human Nav1.7 channel in complexes with incredibly diverse drugs, including bupivacaine, lacosamide, carbamazepine, vinpocetine, hardwickiic acid, and vixotrigine. Structures of drug – hNav1.7 channel complexes revealed new binding sites within and around the pore-forming domain. The authors confirmed binding pose for PF-05089771 in the intact Nav1.7-VSDIV. Remarkably, peptide toxins (previously published structures) and small molecule drugs (structures reported in this manuscript) that bind to unrelated sites lead to similar gate rearrangement involving an α to n helix transition in the middle of the S6IV segment. Based on all currently available structural data, including this manuscript, new nomenclature for binding sites for Nav antagonists is proposed. New hNav1.7 – antagonists structures will be useful for rational design of novel therapeutics targeting Nav channels.

Minor comments:

1. Add to Table 1 drug abbreviations used in the main text.
2. Discuss any available experimental data on hNav1.7 Leu964, Ile1457, and Ile1756 residues (and corresponding residues in homologous Nav channels) that are forming the BIG site for interaction with BPV, LCM, CBZ, and other drugs.
3. Discuss any available experimental data on hNav1.7 residues forming the LCM-2 site (and corresponding residues in homologous Nav channels) with LCM.
4. Please consider adding other key references at the end of the following sentence on lines 212-213 "The corresponding site in Nav channels has been suggested to be targeted by local anesthetics": 1. PMID: 8085162 DOI: 10.1126/science.8085162 ; 2. PMID: 11024055 DOI: 10.1074/jbc.M006992200 ; 3. PMID: 12130650 DOI: 10.1074/jbc.M206126200; 4. PMID: 30728299 DOI: 10.1073/pnas.1817446116.
5. Please consider adding a key reference at the end of the following sentence on lines 288-289 "For example, A-803467, which binds to Site CC4, is a Nav1.8-selective blocker": PMID: 17483457 DOI: 10.1073/pnas.0611364104.

Reviewer #2:

Remarks to the Author:

In this manuscript, Qiurong and colleagues performed a comprehensive analysis of the inhibitory mechanism of Nav1.7 by different antagonist. The high-resolution complex structures of Nav1.7 with antagonists bound revealed unique binding mode for these inhibitors. Structural comparison and analysis identified a common binding site BIG (beneath the intracellular gate), shared by several small molecule drugs including carbamazepine (CBZ), bupivacaine (BPV), and lacosamide (LCM). In addition, the binding modes of several other antagonists, including vinpocetine (VPC), hardwickiic acid (HDA), PF-05089771(PF), and vixotrigine (VXT), have been illustrated at atomic level. These results provided valuable information not only for understanding the inhibitory mechanism of various inhibitors on Nav1.7, but also for future optimizing of drugs targeting Nav1.7.

Furthermore, on the basis of available high-resolution structural information, the authors mapped all binding sites on Nav1.7 and proposed a novel nomenclature system to reflect more accurate binding mode in the 3D space. Foreseeably, this system will set up a way in description of binding mode of small molecules on sodium channel, and be extended to other proteins when enough high-resolution structures are available.

Briefly, this work depict a comprehensive picture of Nav1.7 regulated by various antagonist at atomic level, and will guide future drug optimization to improve specificity and efficiency.

It is interesting that the site BIG is occupied by detergent in apo structure. Have the author observed

the similar density in other structures like Nav1.7/lidocaine, although density for lidocaine is missing? And also for three antagonists share the similar binding site BIG, is it possible to identify unique residues that responsible for difference of these three antagonists in IC50?

1. For picture with several panels, it will be informative to provide more details in figure citation. For example, Consequently, four gating residues, Leu398, ..., move close to tighten the gate (Fig. 1d right panel).
2. Table 1: please explain meaning of different background colors.
3. Fig.1 a. Are density corresponding to small molecules shown in the similar contour level?
4. Fig.1d mid panel. Curves of pore profile are broken.
5. Extended Data Fig. 5: color code for panel D is not clear.
6. Please add in-silico docking simulation in methods section.

Reviewer #1:

This reviewer thinks highly of our work. They only raised several minor concerns that are addressed below.

1. *Add to Table 1 drug abbreviations used in the main text.*

Point taken. Drug abbreviations have been added to improve the clarity of the presentation.

2. *Discuss any available experimental data on hNav1.7 Leu964, Ile1457, and Ile1756 residues (and corresponding residues in homologous Nav channels) that are forming the BIG site for interaction with BPV, LCM, CBZ, and other drugs.*

We thank the reviewer for the thoughtful suggestion. We could not find any experimental data for hNav1.7 Leu964, Ile1457, and Ile1756, or the corresponding ones in other subtypes of Nav channels from literature search. This actually endorses our structural finding of a completely novel drug binding site. Fortunately, the distinct contours of the drug molecules and the excellent map quality cross-validate the assignment of these drugs to site BIG.

3. *Discuss any available experimental data on hNav1.7 residues forming the LCM-2 site (and corresponding residues in homologous Nav channels) with LCM.*

We appreciate this thoughtful suggestion. Despite an exhaustive search through the literature, we were unable to identify any report on the key residues in Nav1.7 responsible for lacosamide binding. Yet we noticed that the hNav1.5-CW/F1760K mutant, which corresponds to F1748 in Nav1.7, displayed complete resistance to lacosamide (Wang, G. K. *et al Molecular pharmacology*, 2014, 85(5): 692-702). Combined with our structural analysis, it appears that F1748 in Nav1.7 plays a critical role in LCM-2 binding through hydrophobic interaction with the phenyl group of LCM-2.

4. *Please consider adding other key references at the end of the following sentence on lines 212-213 “The corresponding site in Nav channels has been suggested to be targeted by local anesthetics”: 1. PMID: 8085162 DOI: 10.1126/science.8085162 ; 2. PMID: 11024055 DOI: 10.1074/jbc.M006992200 ; 3. PMID: 12130650 DOI: 10.1074/jbc.M206126200; 4. PMID:30728299 DOI: 10.1073/pnas.1817446116.*
5. *Please consider adding a key reference at the end of the following sentence on lines 288-289 “For example, A-803467, which binds to Site CC4, is a Nav1.8-selective blocker”: PMID: 17483457 DOI: 10.1073/pnas.0611364104.*

Point taken. Please refer to refs. # 54-57 and ref. # 66 in the revised manuscript.

We thank this reviewer for carefully reading our manuscript and for all the constructive comments.

Reviewer #2:

This reviewer fully recognized the significance of our work and only raised several specific concerns that are addressed below.

1. *It is interesting that the site BIG is occupied by detergent in apo structure. Have the author observed the similar density in other structures like Nav1.7/lidocaine, although density for lidocaine is missing? And also for three antagonists share the similar binding site BIG, is it possible to identify unique residues that responsible for difference of these three antagonists in IC50?*

We appreciate these insightful questions. During data processing for Nav1.7 treated with lidocaine, we did not observe any density similar to the ones that belong to BPV/LCM/CBZ at site BIG. In contrast, an elongated linear density that can be fitted with the detergent molecule glycol-diosgenin (GDN) still penetrates the intracellular gate. In addition, no $\alpha \rightarrow \pi$ transition of the S6_{IV} segment occurs in the Nav1.7-lidocaine dataset, further supporting an apo-state.

The different IC₅₀ values of the three drugs (BPV/LCM/CBZ) are particularly intriguing. At site BIG, they share similar binding poses. However, local coordination cannot provide an immediate clue to their distinct IC₅₀ values, which are further complicated by the 2nd binding site for LCM. Given that site BIG is on the intracellular side, these drugs likely access it by penetrating the membrane in our cell-based electrophysiological measurements. The chemical properties of the drugs may also affect the IC₅₀ values. For instance, BPV is the most hydrophobic among the three antagonists, which may in part account for its lowest IC₅₀ value. Nevertheless, the distinct contours of the drug molecules and the excellent map quality cross-validate the assignment of these drugs to site BIG.

Minor comments:

1. For picture with several panels, it will be informative to provide more details in figure citation. For example, Consequently, four gating residues, Leu398, ..., move close to tighten the gate (Fig. 1d right panel).

Point taken. We have modified the legend for **Figure 1d** to provide a more concise and descriptive explanation for each panel in the revised manuscript “**d**, Further contracted intracellular gate in the presence of the site BIG-binding drugs. Left: Structural comparison of Nav1.7-BPV and Nav1.7-apo (silver). Middle: The pore radii of Nav1.7 bound to different ligands are calculated in HOLE and tabulated. Right: The $\alpha \rightarrow \pi$ transition of S6IV and rearrangement of the gating residues, such as Leu398, Leu964, Ile1547 and Ile1756 (shown as ball and sticks), in the presence of BPV lead to gate contraction. LCM and CBZ-bound structures display nearly identical conformations to that of Nav1.7-BPV.”

2. Table 1: please explain meaning of different background colors.

We are sorry for the confusion. We added “The shade colors code for distinct chemical skeletons of the listed compounds.” to the legend of Table 1 in the revised manuscript.

3. Fig.1 a. Are density corresponding to small molecules shown in the similar contour level?

We are sorry for not including the contour level in the original manuscript. We have added the following information to the corresponding legend:

Figure 1a: The densities for the small molecules are presented at similar levels in Chimera, 5 σ for GDN, 4 σ for BPV, 4 σ for LCM-1, and 4.5 σ for CBZ.

Figure 3a: The contour level for LCM-2 in the left panel is 5 σ .

4. Fig.1d mid panel. Curves of pore profile are broken.

Point taken. In revised **Figure 1d**, we replaced the original data point with a smooth curve to display the HOLE data more clearly.

Fig. 1 | BPV, LCM, and CBZ bind to the same site beneath the intracellular gate (Site BIG).

5. *Extended Data Fig. 5: color code for panel D is not clear.*

Point taken. We have updated the colors of **panel D** in the revised **Supplementary Fig. 5** to enhance the contrast between different structures.

Supplementary Fig. 5 | PF-05089771 binds to Na_v1.7 VSD_{IV}.

6. Please add *in-silico* docking simulation in methods section.

Point taken. We have included the *in silico* molecular docking simulation in the **method** section of our revised manuscript (Page 7 in **SI**).

We thank this reviewer for carefully reading our manuscript and for all the constructive comments.